# A mobile genetic element increases bacterial host fitness by manipulating development

Joshua M Jones[1], Ilana Grinberg[2], Avigdor Eldar[2]*, Alan D Grossman[1]*

[1]Department of Biology, Massachusetts Institute of Technology, Cambridge, United States; [2]The Shmunis School of Biomedicine and Cancer Research, Faculty of Life Sciences, Tel Aviv University, Tel Aviv, Israel

**Abstract** Horizontal gene transfer is a major force in bacterial evolution. Mobile genetic elements are responsible for much of horizontal gene transfer and also carry beneficial cargo genes. Uncovering strategies used by mobile genetic elements to benefit host cells is crucial for understanding their stability and spread in populations. We describe a benefit that ICE*Bs1*, an integrative and conjugative element of *Bacillus subtilis*, provides to its host cells. Activation of ICE*Bs1* conferred a frequency-dependent selective advantage to host cells during two different developmental processes: biofilm formation and sporulation. These benefits were due to inhibition of biofilm-associated gene expression and delayed sporulation by ICE*Bs1*-containing cells, enabling them to exploit their neighbors and grow more prior to development. A single ICE*Bs1* gene, *devI* (formerly *ydcO*), was both necessary and sufficient for inhibition of development. Manipulation of host developmental programs allows ICE*Bs1* to increase host fitness, thereby increasing propagation of the element.

**\*For correspondence:**
avigdor@gmail.com (AE);
adg@mit.edu (ADG)

**Competing interests:** The authors declare that no competing interests exist.

## Introduction

Conjugative elements and phages are abundant mobile genetic elements in bacteria, capable of transferring DNA between cells (*Frost et al., 2005*). Integrative and conjugative elements (ICEs) appear to be the most widespread type of conjugative element (*Guglielmini et al., 2011*). ICEs are found integrated in a host genome. When activated, they excise and produce conjugation machinery that transfers the element DNA from the host cell to recipients (*Carraro and Burrus, 2015*; *Johnson and Grossman, 2015*; *Wozniak and Waldor, 2010*).

ICEs often carry 'cargo' genes that are not necessary for transfer but confer a phenotype to host cells. In fact, ICEs (conjugative transposons) were first identified because of the phenotypes conferred by cargo genes (*Franke and Clewell, 1981*). Cargo genes include those encoding antibiotic resistances, metabolic pathways, and determinants of pathogenesis and symbiosis (*Johnson and Grossman, 2015*). Transfer of mobile elements between cells contributes to rapid evolution and spread of associated cargo genes and phenotypes (*Frost et al., 2005*; *Treangen and Rocha, 2011*).

Despite the benefits cargo genes can provide, the maintenance and transfer of mobile genetic elements requires host cellular resources and in some cases is lethal (*Baltrus, 2013*). Maintenance of a mobile genetic element in host cells requires balancing the costs and benefits to the host or a sufficiently high transfer frequency. Many mobile elements, especially ICEs, have been identified bioinformatically (*Bi et al., 2012*; *Guglielmini et al., 2011*). Many of these ICEs contain putative cargo genes. However, the phenotypes conferred by these genes cannot be inferred from sequence nor are they easily detected experimentally (*Cury et al., 2017*).

ICE*Bs1*, a relatively small (~20 kb) ICE found in most strains of *Bacillus subtilis*, was identified bioinformatically (*Burrus et al., 2002*) and experimentally based on its regulation by cell-cell signaling

**eLife digest** Many bacteria can 'have sex' – that is, they can share their genetic information and trade off segments of DNA. While these mobile genetic elements can be parasites that use the resources of their host to make more of themselves, some carry useful genes which, for example, help bacteria to fight off antibiotics.

Integrative and conjugative elements (or ICEs) are a type of mobile segments that normally stay inside the genetic information of their bacterial host but can sometimes replicate and be pumped out to another cell. ICEBs1 for instance, is an element found in the common soil bacterium *Bacillus subtilis.* Scientists know that ICEBs1 can rapidly spread in biofilms – the slimly, crowded communities where bacteria live tightly connected – but it is still unclear whether it helps or hinders its hosts.

Using genetic manipulations and tracking the survival of different groups of cells, Jones et al. show that carrying ICEBs1 confers an advantage under many conditions. When *B. subtilis* forms biofilms, the presence of the *devI* gene in ICEBs1 helps the cells to delay the production of the costly mucus that keeps bacteria together, allowing the organisms to 'cheat' for a little while and benefit from the tight-knit community without contributing to it. As nutrients become scarce in biofilms, the gene also allows the bacteria to grow for longer before they start to form spores – the dormant bacterial form that can weather difficult conditions.

Mobile elements can carry genes that make bacteria resistant to antibiotics, harmful to humans, or able to use new food sources; they could even be used to artificially introduce genes of interest in these cells. The work by Jones et al. helps to understand the way these elements influence the fate of their host, providing insight into how they could be harnessed for the benefit of human health.

(*Auchtung et al., 2005*). Most of the ICEBs1 genes needed for conjugation are grouped together in an operon that is repressed until activating signals are sensed (*Figure 1*). Two pathways activate ICEBs1, both of which lead to cleavage of the repressor ImmR by the protease and anti-repressor ImmA (*Auchtung et al., 2007*; *Bose et al., 2008*; *Bose and Grossman, 2011*). ICEBs1 contains the cell-cell signaling genes, *rapI* and *phrI*, which regulate ICEBs1 activation by sensing population density and the relative abundance of ICEBs1-containing host cells (*Auchtung et al., 2005*). RapI is produced at high cell density and during the transition to stationary phase and stimulates the proteolytic cleavage of the repressor ImmR by the protease ImmA (*Bose and Grossman, 2011*). Overproduction of RapI stimulates activation of ICEBs1 in >90% of cells (*Auchtung et al., 2005*). RapI activity (and therefore ICEBs1 activation) is inhibited by PhrI, a peptide that is secreted by cells that contain ICEBs1. PhrI levels indicate the relative abundance of ICEBs1-containing cells in the population, preventing the activation and possible redundant transfer of ICEBs1 if most nearby cells already contain the element. ICEBs1 is also activated during the RecA-dependent DNA damage response (*Auchtung et al., 2005*).

Biofilms appear to be hotspots of horizontal gene transfer for bacteria growing in natural settings (*Madsen et al., 2012*; *Molin and Tolker-Nielsen, 2003*). Undomesticated strains of *B. subtilis* form complex biofilms on agar plates and at the air-liquid interface in standing cultures (*Vlamakis et al., 2013*). There is also extensive spore formation in *B. subtilis* biofilms (*Branda et al., 2001*; *Vlamakis et al., 2008*). In addition, during growth in a biofilm, ICEBs1 is naturally activated and transfers efficiently, generating on the order of 10 new ICEBs1-containing host cells (transconjugants) per donor cell under appropriate conditions (*Lécuyer et al., 2018*). *B. subtilis* biofilms are held together by a matrix composed of secreted exopolysaccharides, protein fibers, and DNA (*Vlamakis et al., 2013*). This matrix reinforces cell-cell contacts, likely promoting rapid spread of ICEBs1 by conjugation. Additionally, the conditions that promote biofilm formation (high cell density) also promote activation and transfer of ICEBs1 and sporulation (*Auchtung et al., 2005*; *Grossman and Losick, 1988*). Although biofilm growth is clearly beneficial to conjugation, it is unknown how ICEBs1 impacts its host cells under these conditions.

In this study, we describe a selective advantage provided by ICEBs1 to its host cells during growth in biofilms. This fitness benefit was due to inhibition of host biofilm and spore development. We identified the ICEBs1 gene *devI* (formerly *ydcO*) as necessary and sufficient to inhibit host development and provide a selective advantage to ICEBs1-containing cells. We also provide evidence

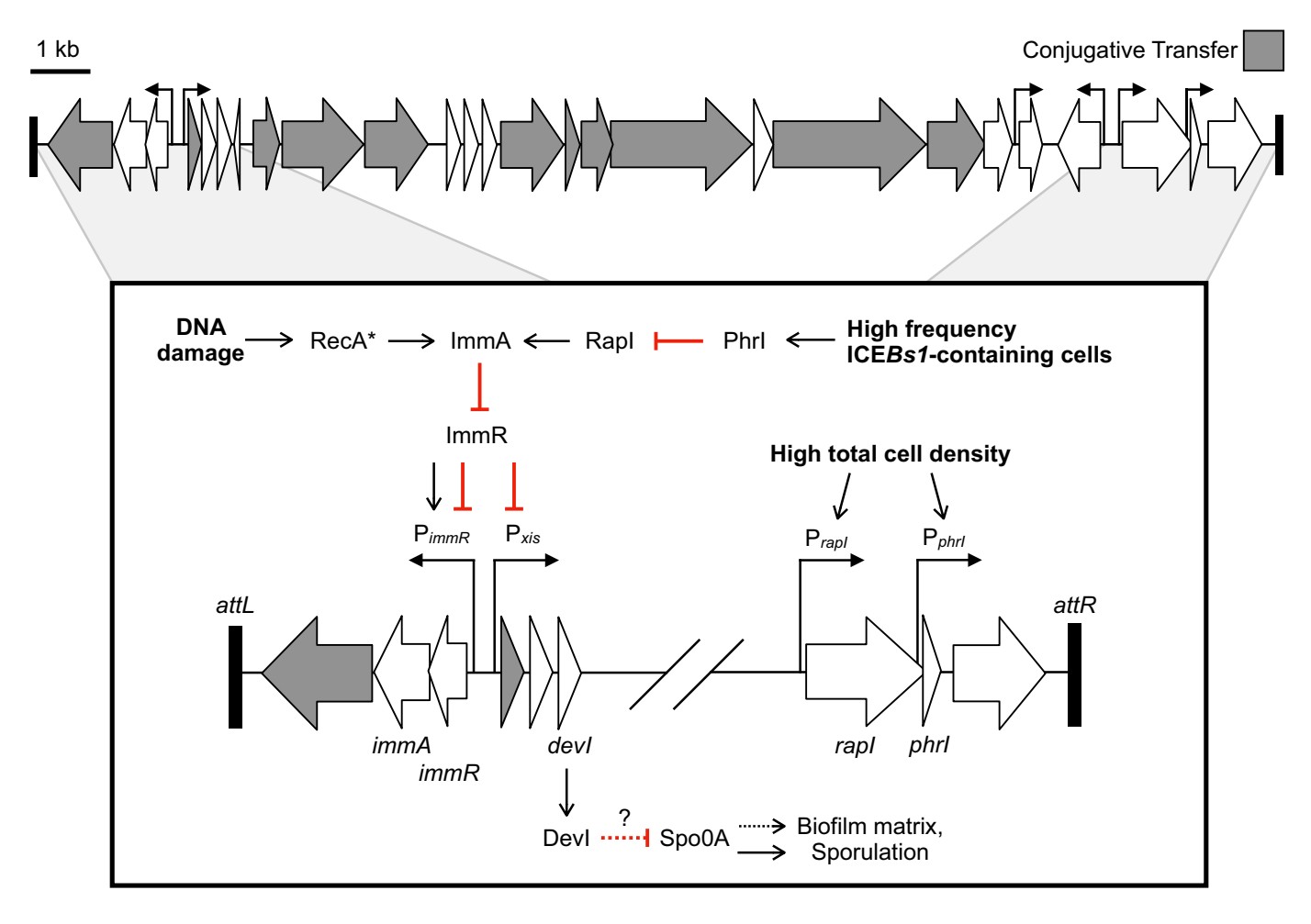

**Figure 1.** Genetic map and regulatory pathways of ICE*Bs1*. Genes are represented by horizontal block arrows indicating the direction of transcription. Vertical right-angle arrows mark the positions of promoters, and the arrowhead indicates the direction of transcription. Genes known to be involved in the conjugative life cycle of ICE*Bs1* are shaded in gray. The 60 bp direct repeats that mark the ends of ICE*Bs1* are shown as black rectangles. (Inset) A partial genetic map that highlights factors involved in the regulation of ICE*Bs1*. The major promoter Pxis drives expression of most genes in ICE*Bs1*. Pxis is repressed by the ICE-encoded repressor ImmR. Repression is relieved when ImmR is cleaved by the protease ImmA, and proteolytic cleavage is stimulated by activated RecA (RecA*) in response to DNA damage, or, independently by the cell signaling regulator RapI. RapI is made when cells are crowded by potential recipients, but repressed by the ICE-encoded secreted peptide PhrI if the neighboring cells already contain a copy of ICE*Bs1*. *devI* (formerly *ydcO*) is the third open-reading frame downstream of Pxis. DevI inhibits sporulation and expression of biofilm matrix genes, likely by inhibiting Spo0A (directly or indirectly). In the genetic pathways, black arrows indicate activation and red T-bars indicate inhibition.

indicating that *devI* likely inhibits the key developmental transcription factor Spo0A, reducing its ability to stimulate biofilm and sporulation gene expression. *devI* (*ydcO*) is conserved in other ICE*Bs1*-like elements, indicating that manipulation of host development may be a conserved strategy among this family of mobile genetic elements. We postulate that manipulation of host pathways may be a common function of many of the as yet uncharacterized cargo genes in ICEs.

## Results

### ICE*Bs1* spreads efficiently in biofilms by conjugation

Biofilm formation is characteristic of many bacteria growing in natural settings, including *B. subtilis*. We used biofilm growth to determine if ICE*Bs1* affected the fitness of its host cells under conditions

that naturally promote its spread. We performed competition experiments in biofilms using strains of undomesticated *B. subtilis* (NCIB3610 plasmid-free) with or without ICE*Bs1*.

We observed highly efficient spread of ICE*Bs1* at low donor to recipient ratios (*Table 1*) during growth in biofilms, similar to results reported previously (*Lécuyer et al., 2018*). To measure mating, we mixed ICE*Bs1*-containing cells (potential donors) with cells that did not contain ICE*Bs1* (ICE*Bs1*$^0$, potential recipients) and co-cultured the mix on standard biofilm-stimulating growth medium (MSgg agar) (*Branda et al., 2001*). Since ICE*Bs1* induction is regulated by cell-cell signaling, we varied the initial frequency of ICE*Bs1*+ cells between approximately 0.01 and 0.9. We inserted unique select-able markers (antibiotic resistances) in the chromosomes of the donors and recipients as well as within ICE*Bs1*. After 4 days of growth at 30°C (approximately 17 net doublings of the initial popula-tion), biofilms were disrupted and the number of transconjugants was determined by selective plating.

We found that after four days of biofilm growth, the frequency of transconjugants ranged from ~0.4 to 0.6 of total cells in the biofilm for starting donor frequencies of ~0.01–0.5 (*Table 1*). The highest frequency of transconjugants was observed when the starting frequency of ICE*Bs1*-con-taining cells was ~0.1. Enhanced conjugation at low donor to recipient ratios is likely due to regula-tion of ICE*Bs1* by cell-cell signaling (induction is inhibited by the presence of other potential donors) and the higher likelihood of contacting potential recipients at low frequencies of donors.

The high levels of ICE*Bs1* conjugation during growth in biofilms presented a challenge for quanti-fying the fitness of ICE*Bs1*-containing host cells relative to ICE*Bs1*$^0$ cells. Mating converts a large fraction of ICE*Bs1*$^0$ cells to transconjugants (ICE*Bs1*-containing), reducing the ICE*Bs1*$^0$ proportion of the population in a manner unrelated to host fitness. To measure the effect of ICE*Bs1* on host fit-ness, we blocked conjugative DNA transfer using the *conEK476E* mutation (*Berkmen et al., 2010*). We then compared the proportion of ICE*Bs1*-containing hosts to ICE*Bs1*$^0$ cells without the con-founding influence of conjugation.

## ICE*Bs1* provides a frequency-dependent selective advantage in biofilms

We found that cells containing ICE*Bs1* that is incapable of conjugation {ICE*Bs1*(*conEK476E*)} had a fitness advantage over cells lacking ICE*Bs1* during biofilm growth when they were initially present as a minority of the population. As before, we varied the initial frequency of ICE*Bs1*-containing host cells in the inoculum between approximately 0.01 and 0.9. To measure fitness we determined the frequency of ICE*Bs1*-containing cells ($f_{ICE}$) and ICE*Bs1*$^0$ ($f_{NULL}$) cells in the initial inoculum and in mature biofilms (4 days of growth at 30°C) by selective plating. The relative fitness of the ICE*Bs1*-containing cells was calculated as the fold change in the ratio of $f_{ICE}$ / $f_{NULL}$ over the course of the competition.

We found that the fitness of ICE*Bs1*-containing cells was dependent on their initial frequency in the population (*Figure 2A*). The frequency-dependence was most likely due to regulation of ICE*Bs1* gene expression by the cell-cell signaling genes *rapI-phrI* or some other function of *rapI*. Cells with ICE*Bs1* had a selective advantage at low frequencies (0.01 or 0.1) when the element is most strongly activated. At high frequencies in the population (0.5 or 0.9), when there is little or no activation,

**Table 1.** Frequency of transconjugants generated in biofilm matings.

| Initial frequency ICE*Bs1* donors[*] | Final frequency transconjugants[†] |
|---|---|
| 0.008 ± 0.002 | 0.42 ± 0.13 |
| 0.10 ± 0.03 | 0.64 ± 0.15 |
| 0.47 ± 0.05 | 0.44 ± 0.07 |
| 0.89 ± 0.03 | 0.063 ± 0.008 |

[*]ICE*Bs1*-containing cells (JMJ592) were mixed with ICE*Bs1*-cured cells (JMJ550). The initial frequencies reported are the average ± standard deviation from three independent experiments.

[†]The final frequencies of transconjugants reported are the average ± standard deviation from a total of nine biofilms from three independent experiments.

The online version of this article includes the following source data for Table 1:

**Source data 1.** Mating in biofilms.Counts of donors, recipients, and transconjugants for ICE*Bs1* mating in biofilms.

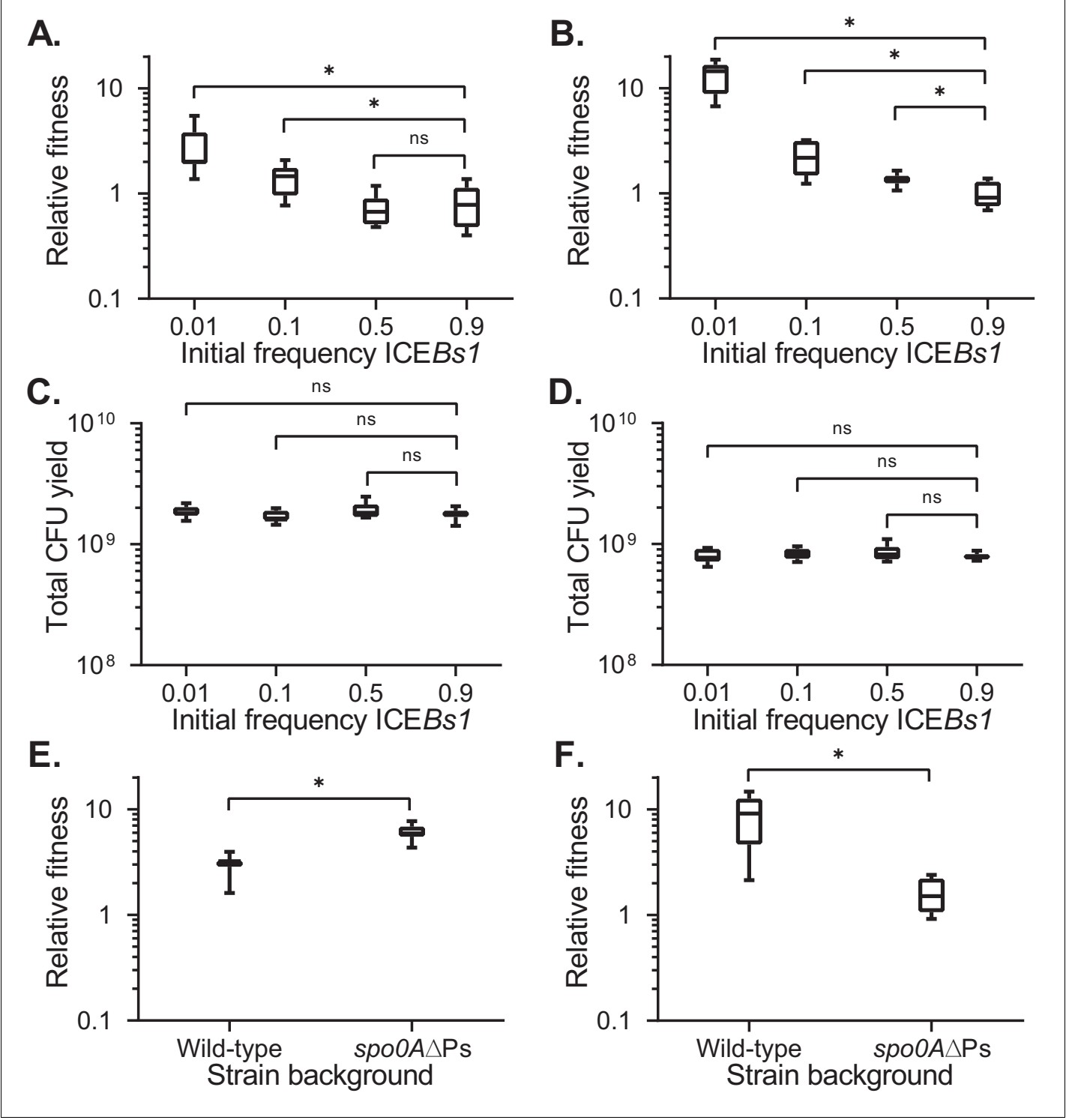

**Figure 2.** Fitness of ICE*Bs1*-containing cells relative to ICE*Bs1*-cured cells during development. The fitness of ICE*Bs1*-containing cells (JMJ593) relative to ICE*Bs1*-cured cells (JMJ550) was measured by competition during biofilm growth (**A**) or growth on sporulation medium (**B**). Total growth yields in biofilms (**C**) and on sporulation medium (**D**) were determined by counting CFUs derived from both cells and spores. The fitness of ICE*Bs1*-containing cells relative to ICE*Bs1*-cured cells was compared between the wild-type strain background (JMJ593 vs. JMJ550) and in the sporulation-deficient *spo0A*ΔPs background (JMJ788 vs. JMJ786) by competition during biofilm growth (**E**) or growth on sporulation medium (**F**). ICE*Bs1*-containing cells were inoculated at an initial frequency of 0.01. Data shown are pooled from three independent experiments. A total of nine populations (biological replicates) were analyzed per condition. Boxes extend from the lower to upper quartiles of the data, and the middle line indicates the median fitness.
*Figure 2 continued on next page*

*Figure 2 continued*

Whiskers indicate the range of the fitness measurements. Asterisks indicate a p-value<0.05 (two-tailed T-test, unequal variance). Exact p-values: (**A**) 3.7 $\times$ 10$^{-5}$, 8.8 $\times$ 10$^{-3}$, 6.0 $\times$ 10$^{-1}$; (**B**) 4.9 $\times$ 10$^{-11}$, 1.6 $\times$ 10$^{-4}$, 7.1 $\times$ 10$^{-3}$; (**C**) 3.7 $\times$ 10$^{-1}$, 4.2 $\times$ 10$^{-1}$, 2.2 $\times$ 10$^{-2}$; (**D**) 8.9 $\times$ 10$^{-1}$, 1.9 $\times$ 10$^{-1}$, 1.7 $\times$ 10$^{-1}$; (**E**) 5.5 $\times$ 10$^{-6}$; (**F**) 2.4 $\times$ 10$^{-5}$.

The online version of this article includes the following source data for figure 2:

**Source data 1.** Frequency-dependent fitness.
**Source data 2.** Fitness dependence on sporulation.

fitness of ICE*Bs1*-containing cells was approximately neutral (*Figure 2A*). The final growth yields of the populations were similar regardless of the frequency of ICE*Bs1*-containing cells (*Figure 2C*).

We performed control competitions of two differentially marked ICE*Bs1*$^0$ strains to verify that the enhanced fitness we observed was due to the presence of ICE*Bs1* rather than an inherent fitness difference associated with antibiotic resistances (see Materials and methods, *Source data 1*). There was a small cost associated with the marker used to select cells containing ICE*Bs1* (median relative fitness 0.7 ± 0.09), leading to a slight underestimate of the selective advantage to these cells.

There is a large amount of sporulation in *B. subtilis* biofilms (*Branda et al., 2001*; *Vlamakis et al., 2008*). Consistent with this, we found that approximately 80% of viable colony-forming units (CFUs) in a mature biofilm after 3 days were from spores. The selective advantage to cells containing ICE*Bs1* growing in biofilms could be due to sporulation and/or biofilm development.

## ICE*Bs1* confers a selective advantage in biofilms without sporulation

We blocked sporulation using a mutation that causes a reduction in the amount of the transcription factor Spo0A that is required for spore formation. The *spo0AΔPs* mutation is a deletion of the sigma-H-dependent promoter upstream of *spo0A* (*Siranosian and Grossman, 1994*). This mutation reduces production of Spo0A, and cells do not achieve the threshold concentration required to initiate sporulation (*Chung et al., 1994*). *spo0AΔPs* mutant cells formed biofilms that were morphologically similar to those formed by wild-type cells.

In biofilms without sporulation, *spo0AΔPs* mutant cells containing ICE*Bs1* (JMJ788) had a selective advantage compared to *spo0AΔPs* mutant cells without ICE*Bs1* (JMJ786) (*Figure 2E*). Notably, the median fitness for *spo0AΔPs* mutant cells containing ICE*Bs1* at a low frequency in the population was approximately six. Thus, sporulation was not required for a fitness benefit to ICE*Bs1*-containing cells in biofilms.

Fitness of cells in biofilms can be affected by production of the biofilm matrix. For example, cells that 'cheat' by contributing less to biofilm matrix production reap the benefits of growing with other cells that bear the cost of matrix gene expression (*Dragoš et al., 2018*). We showed that cells containing ICE*Bs1* 'cheat' by decreasing expression of biofilm matrix genes compared to cells without ICE*Bs1* (see below).

## ICE*Bs1* confers a selective advantage during sporulation, in the absence of biofilms

We found that cells containing ICE*Bs1* also had a frequency-dependent selective advantage during sporulation, in the absence of biofilms. We prepared mixtures of cells with and without ICE*Bs1* as described above. These mixtures were spotted onto a medium (DSM agar) that promotes high levels of sporulation. During growth on this medium, there are no complex colony features found in biofilms. As in the biofilm competitions, cells containing ICE*Bs1* had a frequency-dependent selective advantage during sporulation (*Figure 2B*). At an initial frequency of approximately 0.01, the median relative fitness of the ICE*Bs1*-containing cells was approximately 14 (14.5 ± 4.3). As in biofilms, the total growth yields of the populations were similar regardless of ICE*Bs1* host frequency (*Figure 2D*). These results demonstrate that ICE*Bs1* confers a selective advantage to cells growing on DSM agar, outside the context of biofilms. This could be due to sporulation or growth under these specific conditions.

We found that sporulation was required for the strong selective advantage during growth on sporulation medium (DSM agar). The fitness benefit associated with the ICE*Bs1*-containing cells at a low frequency in the population was greatly reduced in the *spo0AΔPs* mutant (no sporulation)

(*Figure 2F*). The sporulation mutant with ICE*Bs1* had a median fitness of approximately 1.5 compared to approximately nine for wild-type. Based on these results, we conclude that the presence of ICE*Bs1* confers a frequency-dependent selective advantage during sporulation.

Together, our results demonstrate that cells containing ICE*Bs1* have a frequency-dependent selective advantage in biofilms and during sporulation. This selective advantage is independent of the ability of ICE*Bs1* to actually transfer from one cell to another. Biofilm formation (*Hamon and Lazazzera, 2001*) and sporulation (*Hoch, 1993*; *Sonenshein, 2000*) are both regulated by the transcription factor Spo0A. Our results indicate that the presence of ICE*Bs1* could somehow be inhibiting the activity or activation of Spo0A.

## ICE*Bs1*-containing cells have a frequency-dependent delay in sporulation

We hypothesized that some ICE*Bs1*-encoded gene(s) inhibit host cell development. This inhibition could delay development and enable cells to continue growth for a small number of generations. This model derives from analogous phenotypes of mutants that do not enter the sporulation pathway (*Dawes and Mandelstam, 1970*). Mutants that delay the start of sporulation have a growth advantage as they are able to divide one or a few more times while other cells in the population stop growing and start to sporulate.

We found that in mixed populations, sporulation was delayed in cells containing ICE*Bs1* compared to cells without ICE*Bs1*, in a frequency-dependent manner. As above, we used an ICE*Bs1* mutant that is incapable of conjugation {ICE*Bs1*(*conEK476E*)}. We started several replicate populations, each of which we sampled once at different times to create a time-course. (This was done because sampling to quantify CFUs [spores and cells] disrupts and prevents monitoring a single population over time.) Spore frequency was determined by measuring heat-resistant CFUs as a fraction of total CFUs for ICE*Bs1*-containing and ICE*Bs1*-cured strains that contained different antibiotic resistance markers to distinguish the strains.

### Sporulation is delayed in ICE*Bs1* host cells during biofilm formation

We found that sporulation of ICE*Bs1*-containing cells was delayed in a frequency-dependent manner during growth in biofilms (*Figure 3A and B*). When ICE*Bs1*-containing cells were started at a frequency of approximately 0.01, they reached their maximum sporulation frequency (>80% spores) roughly 17 hr later than the cells without ICE*Bs1* (*Figure 3A*). After 3 days of biofilm growth, the sporulation frequencies of ICE*Bs1*-containing and ICE*Bs1*-cured cells were indistinguishable. Over this period of time the total frequency of ICE*Bs1*-containing cells in the population typically rose from ~0.01 to ~0.03, giving a relative fitness (~3) consistent with results above (*Figure 2A*). When the ICE*Bs1*-containing cells were the majority in the population (initial frequency ~0.9) the timing and sporulation frequencies of the ICE*Bs1*-containing and ICE*Bs1*-cured cells were indistinguishable (*Figure 3B*).

### Sporulation is delayed in ICE*Bs1* host cells during sporulation in the absence of biofilms

We also found that sporulation of ICE*Bs1*-containing cells was delayed in a frequency-dependent manner during sporulation in the absence of biofilm formation (*Figure 3C and D*). When the ICE*Bs1* containing cells were inoculated at a low frequency (approximately 0.01), the delay in sporulation was qualitatively similar to that observed in biofilms (*Figure 3C*). However, the increase in the total frequency of ICE*Bs1*-containing cells in the population was approximately 10-fold, giving a relative fitness of approximately 10, consistent with results described above (*Figure 2B*). This increase was much greater than the approximately threefold increase during biofilm formation.

We suspect that the stronger selective advantage of ICE*Bs1*-containing cells on sporulation medium is due to the earlier onset of sporulation. By 16 hr of growth on sporulation medium, spores made up about 40% of the total CFUs. By the same time in biofilms, spores were undetectable (limit of detection ~0.03% spores).

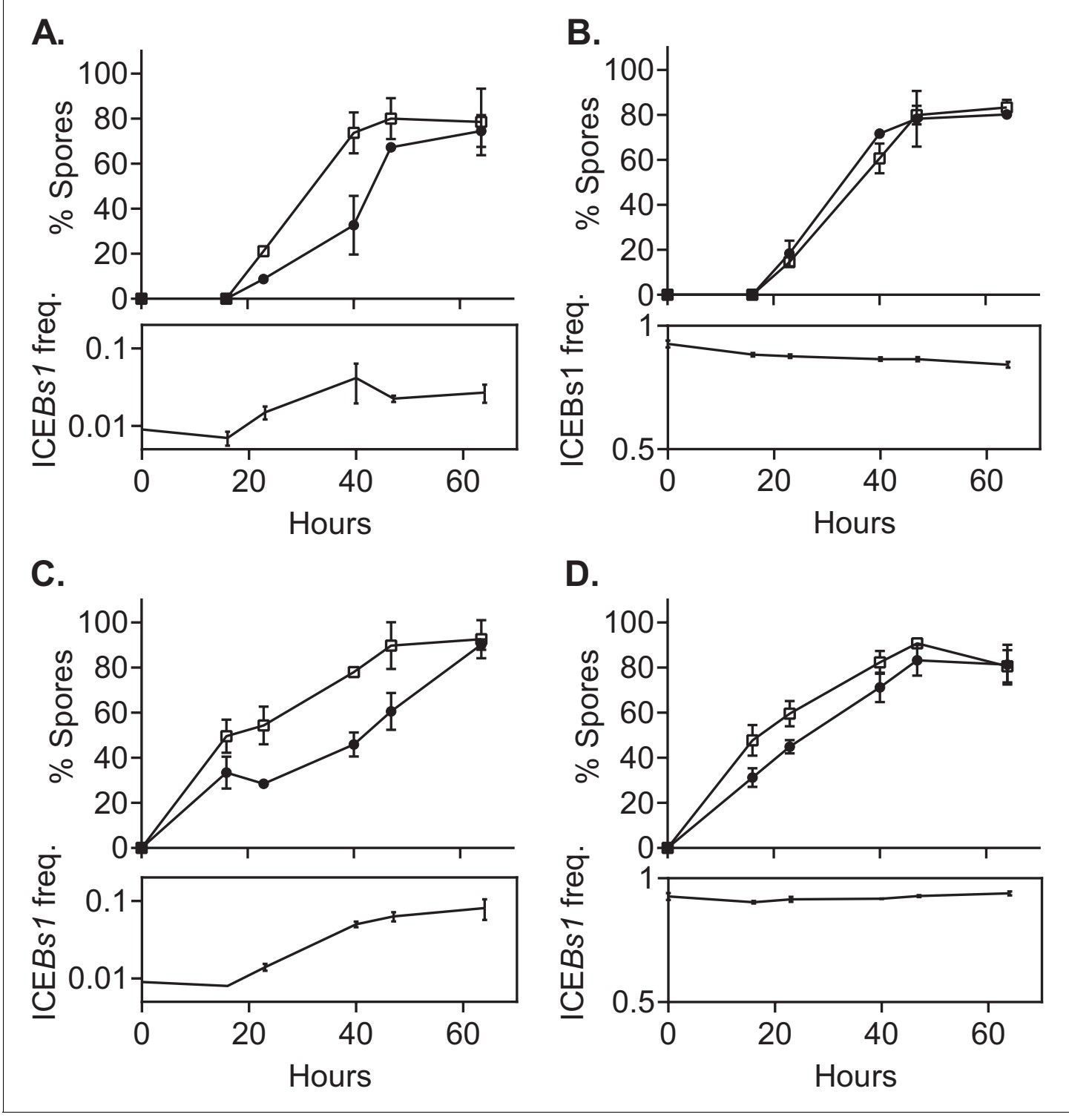

**Figure 3.** ICE*Bs1*-containing cells delay sporulation in a frequency-dependent manner. ICE*Bs1*-containing cells (JMJ593, black circles) were mixed with ICE*Bs1*-cured cells (JMJ550, open squares) at an initial frequency of 0.01 and spotted onto biofilm growth medium (**A**) or sporulation medium (**C**). Mixtures with ICE*Bs1*-containing cells at an initial frequency of 0.9 were also spotted onto biofilm medium (**B**) and sporulation medium (**D**). Biofilms and colonies were harvested at the indicated times to determine the fraction of CFUs derived from heat-resistant spores for both the ICE*Bs1*-containing and the ICE*Bs1*-cured cells. Boxes below each graph indicate the frequency of ICE*Bs1*-containing CFUs at each timepoint. Data shown are the average from two populations (biological replicates) per timepoint with error bars indicating the standard deviation. A representative experiment is shown.

The online version of this article includes the following source data for figure 3:

**Source data 1.** Sporulation timing during competitions.

### *rapI-phrI* are necessary but not sufficient for enhanced fitness

The fitness benefits provided by ICE*Bs1* were dependent on the relative abundance of ICE*Bs1*-containing cells, indicating that the cell-cell signaling genes *rapI-phrI* in ICE*Bs1* were likely involved, either directly or indirectly. Other Rap proteins in *B. subtilis* are known to regulate development by inhibiting phosphorylation (activation) of the transcription factor Spo0A (*Sonenshein, 2000*). RapI, like other Rap proteins in *B. subtilis*, can inhibit the pathway needed to phosphorylate (activate) the transcription factor Spo0A, and overexpression of *rapI* in vivo inhibits sporulation (*Even-Tov et al., 2016*; *Parashar et al., 2013*; *Singh et al., 2013*). However, it was unknown whether RapI regulates development in vivo under physiological conditions. Results described below demonstrate that the *rapI-phrI* system is required for the fitness advantage of ICE*Bs1*-containing cells, but that this requirement is by virtue of causing induction of ICE*Bs1* gene expression. Another gene in ICE*Bs1* is both necessary and sufficient for the selective advantage of ICE*Bs1*-containing cells during development.

#### *rapI-phrI* are required for the fitness benefit of cells containing ICE*Bs1*

We deleted *rapI-phrI* (Δ*rapI-phrI*) in ICE*Bs1* and compared the fitness conferred by this mutant to that conferred by ICE*Bs1* with *rapI-phrI*. Because loss of *rapI* prevents induction of gene expression, excision, and replication of ICE*Bs1*, we used ICE*Bs1* mutants ('locked-in') that are incapable of excision or replication (see Materials and methods), regardless of the presence or absence of *rapI*. Preventing excision and replication of ICE*Bs1* allowed us to compare the fitness of wild-type ICE*Bs1* to ICE*Bs1* Δ*rapI-phrI* (and other mutants), which would otherwise have a lower gene copy number due to a lower frequency of induction.

We verified that locked-in ICE*Bs1* still conferred a fitness benefit to host cells. During sporulation in biofilms (MSgg agar), cells containing locked-in ICE*Bs1* had a relative fitness of approximately 14 when they were started at a low frequency in the population (~0.01) (*Figure 4A*). This benefit was much greater than that conferred by wild-type ICE*Bs1* that can excise and replicate. We suspect that replication of ICE*Bs1* incurs a fitness cost to the host cell that reduces the apparent benefit. The sources of this burden could include use of host resources, the host's response to single-stranded DNA produced by rolling-circle-replication of ICE*Bs1*, and increases in ICE*Bs1* gene expression due to increased copy number.

We found that *rapI-phrI* were required for the fitness benefit conferred by ICE*Bs1*. During sporulation in biofilms (MSgg agar), the relative fitness of the Δ(*rapI-phrI*) host strain was approximately neutral (*Figure 4A*), in contrast to the high fitness (median ~14) provided by ICE*Bs1* containing *rapI-phrI* when the ICE*Bs1*-containing cells were started at a low frequency (~0.01). The requirement for *rapI-phrI* could be due to a direct role for one of these, likely RapI, or an indirect role in activating expression of ICE genes.

#### *rapI-phrI* are not sufficient in the absence of other ICE*Bs1* genes to provide a fitness benefit

We found that *rapI-phrI* alone were not sufficient to provide a fitness benefit during sporulation or during sporulation in biofilms. We cloned *rapI-phrI* and their native promoters and inserted them in an ectopic locus (*bcaP*) in a strain that did not contain ICE*Bs1*. When this strain was started at a low frequency (~0.01), fitness of this strain was neutral relative to a control strain without *rapI-phrI* (*Figure 4B*). To verify that *rapI-phrI* were functional, we added back ICE*Bs1* that was missing *rapI-phrI*. Adding the rest of ICE*Bs1* restored the fitness advantage during sporulation and in biofilms, indicating that the ectopic copy of *rapI-phrI* was functional (*Figure 4B*). The requirement for *rapI-phrI* and some other ICE*Bs1* gene(s) indicated that the selective advantage was likely dependent on induction of ICE*Bs1* by RapI.

#### Activation of ICE*Bs1* is required for the fitness benefit

We found that expression of one or more ICE*Bs1* genes controlled by the promoter P*xis* was required for the selective advantage in biofilms with sporulation. P*xis* drives most of the genes in ICE*Bs1* and is indirectly activated by RapI in a frequency-dependent manner (*Auchtung et al., 2005*; *Bose and Grossman, 2011*). We deleted P*xis* in a strain in which ICE*Bs1* was unable to excise or replicate (locked-in-ICE*Bs1*). In this strain, only genes not dependent on P*xis* could be expressed,

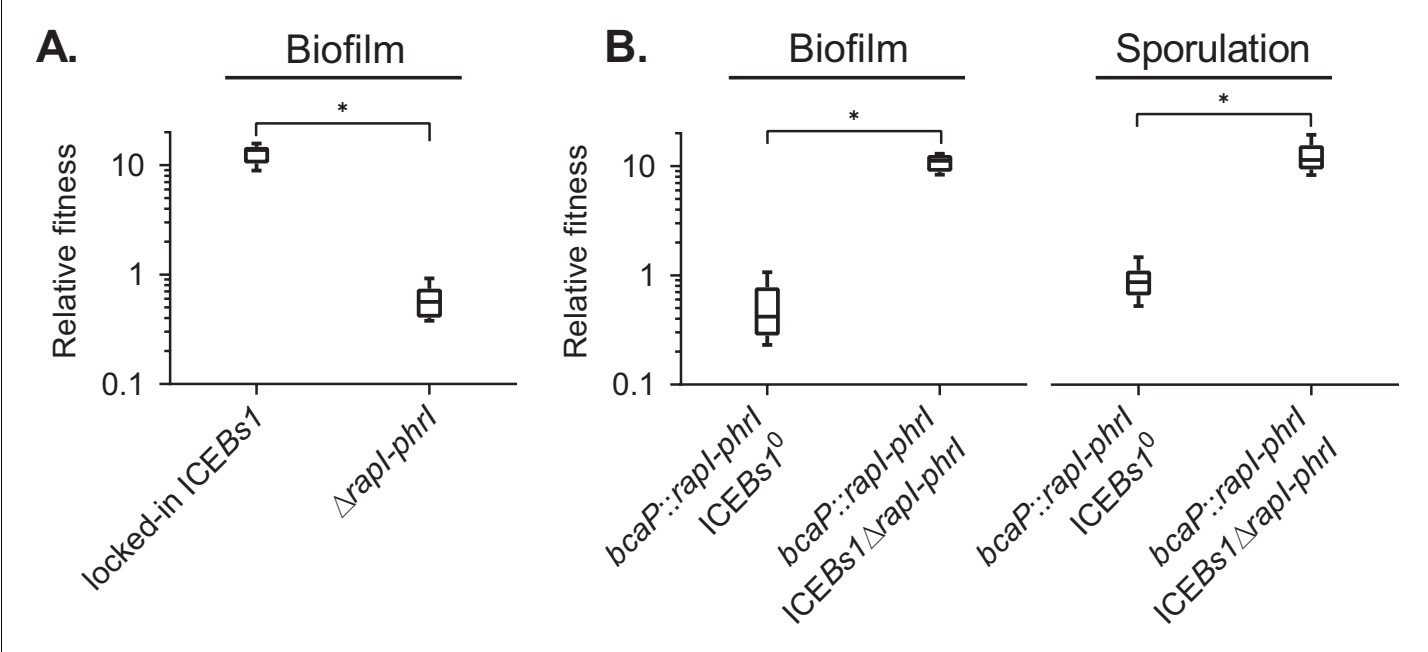

**Figure 4.** The ICE*Bs1* cell-cell signaling genes, *rapI-phrI*, are necessary but not sufficient to confer a selective advantage. (**A**) The fitness of cells containing locked-in ICE*Bs1* (JMJ646) or an isogenic *rapI-phrI* mutant (JMJ686) was measured relative to ICE*Bs1*-cured cells (JMJ550) during biofilm competitions. The ICE*Bs1*-containing cells were started at a frequency of 0.01. (**B**) The fitness of cells containing *rapI-phrI* alone (JMJ576) or cells containing both *rapI-phrI* and locked-in ICE*Bs1*Δ*rapI-phrI* (JMJ785) was measured relative to ICE*Bs1*-cured cells (JMJ714) during biofilm and sporulation medium competitions. JMJ576 and JMJ785 were started at a frequency of 0.01. Data shown are pooled from three independent experiments with a total of nine populations (biological replicates) analyzed per strain mixture. Boxes extend from the lower to upper quartiles, and the middle line indicates the median fitness. Whiskers indicate the range of the fitness measurements. Asterisks indicate a p-value<0.05 (two-tailed T-test, unequal variance). Exact p-values: (**A**) $1.3 \times 10^{-12}$; (**B**) $2.1 \times 10^{-8}$, $4.6 \times 10^{-2}$.

The online version of this article includes the following source data for figure 4:

**Source data 1.** Fitness dependence on *rapI-phrI*.

including *rapI-phrI*. Fitness of this strain was approximately neutral during sporulation in biofilms (*Figure 5B*). This indicated that expression of one or more genes controlled by P*xis*, either alone or in combination with *rapI*, was required for the fitness benefit conferred by ICE*Bs1*.

Since most of the genes controlled by P*xis* have known roles in the conjugative life cycle, we focused our search on genes without a known function, starting with the genes near P*xis*. Of these, we found that a deletion of *devI* (*ydcO*) reduced fitness. Results described below demonstrate that a single ICE*Bs1* gene, *devI* (*ydcO*), is both necessary and sufficient to inhibit development and provide a selective advantage. The primary role of *rapI* in the fitness benefit appears to be the induction of *devI* expression.

### *devI* is necessary for the fitness benefit conferred by ICE*Bs1*

We found that an ICE*Bs1* gene of unknown function, *devI* (*ydcO*), was necessary for the fitness advantage of ICE*Bs1* host cells. We constructed a deletion of *devI* in the locked-in-ICE*Bs1* strain. When started at a low frequency in the population (~0.01) the relative fitness of the *devI* mutant was approximately 3.5 (*Figure 5D*), much less than that of the isogenic *devI*+ cells (median fitness ~14) in biofilms with sporulation (*Figure 5A*). Interestingly, the deletion of *devI* did not reduce fitness fully to neutral, indicating a possible role for other ICE*Bs1* genes. *devI* (*ydcO*) is predicted to encode an 86 amino acid protein. A search for conserved motifs and structural similarity between DevI (YdcO) and other proteins did not significantly inform our understanding of DevI function. However, *devI* (*ydcO*) homologs are found in other *Bacillus* species (see below).

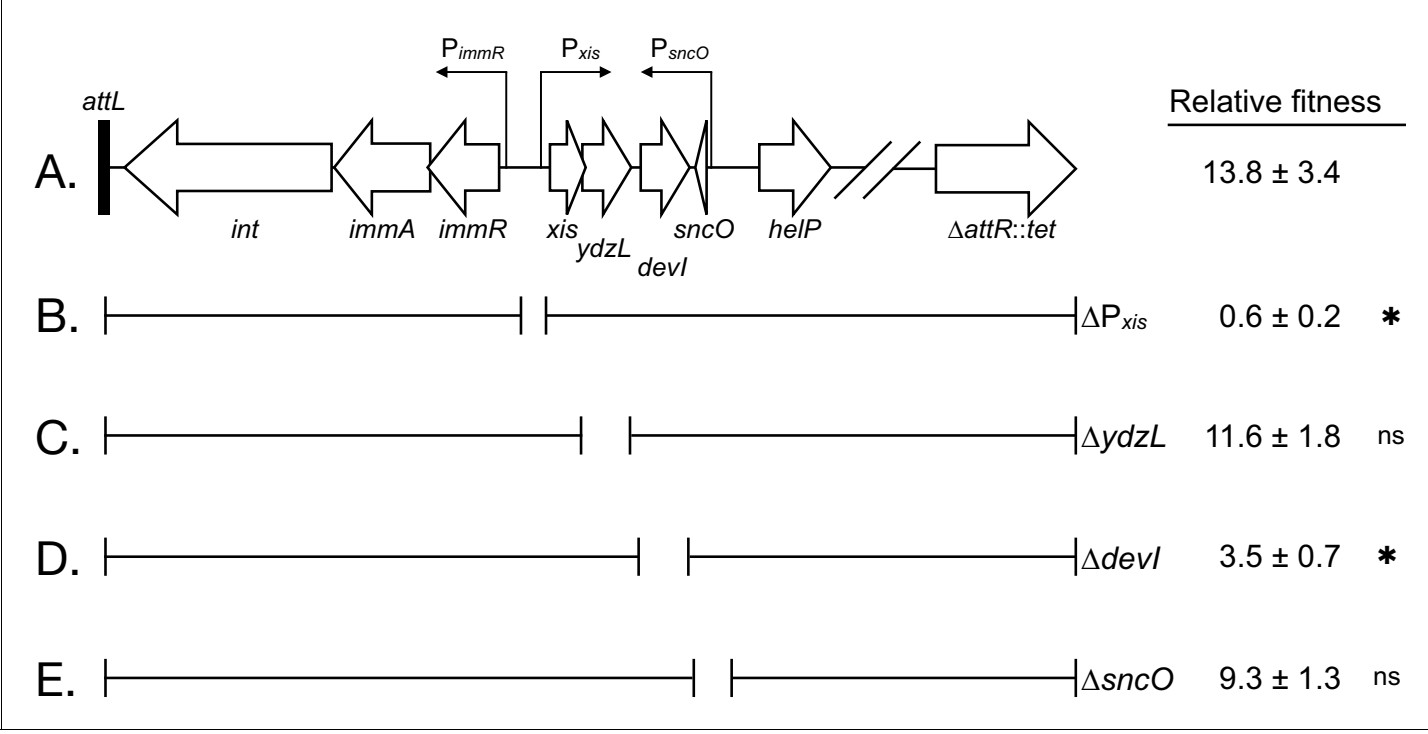

**Figure 5.** Effects of deletions in ICE*Bs1* on host fitness. (**A**) Abbreviated genetic map of locked-in-ICE*Bs1* showing genes as open block arrows, promoters as thin right-angle arrows, and the left attachment site (*attL*) as a black bar. (**B** to **E**) Brackets under the map of ICE*Bs1* indicate regions contained in isogenic derivatives of locked-in-ICE*Bs1*. Open spaces represent regions deleted. The fitness of strains containing locked-in-ICE*Bs1* (JMJ646) and its derivatives (ΔP*xis*, JMJ662; Δ*ydzL*, JMJ704; Δ*devI*, JMJ703; Δ*sncO*, JMJ688) is indicated at the right. Fitness was measured relative to ICE*Bs1*-cured cells (JMJ550) during competitions in biofilms. ICE*Bs1*-containing cells were started at a frequency of 0.01. Data shown are the median ± standard deviation from at least three independent experiments (at least nine total biological replicates per strain mixture), with the exception of JMJ688 which was measured only in one experiment (three total biological replicates). Asterisks indicate a statistically significant difference in fitness compared to JMJ646 (p-value<0.05, two-tailed T-test, unequal variance). Exact p-values: (**B**) $2.3 \times 10^{-16}$; (**C**) $4.5 \times 10^{-1}$; (**D**) $6.3 \times 10^{-12}$; (**E**) $6.2 \times 10^{-2}$. The online version of this article includes the following source data for figure 5:

**Source data 1.** Effects of gene deletions on ICE*Bs1* fitness.

### *devI* is sufficient to inhibit sporulation and provide a fitness benefit

We found that when expressed constitutively, *devI* alone, in the absence of all other ICE*Bs1* genes, was sufficient to inhibit sporulation and provide a selective advantage. We cloned *devI* under the control of P*xis* at an ectopic locus (*lacA*) in a strain without ICE*Bs1*. In the absence of ICE*Bs1* (and its repressor ImmR), P*xis* is constitutively active (*Auchtung et al., 2007*). Fitness was measured relative to a control strain that had P*xis* with no gene downstream.

Sporulation of the P*xis-devI* strain was strongly inhibited under conditions that normally support robust sporulation, including in biofilms (*Figure 6*). During sporulation either with (*Figure 6A*) or without biofilm formation (*Figure 6B*), the frequency of the P*xis-devI* strain in the population rose from ~0.01 to ~0.05, giving a relative fitness of ~5. This is greater than the typical fitness conferred by ICE*Bs1* in biofilms, but less than that observed during sporulation without biofilms. We suspect these differences are due to constitutive expression of *devI* in the absence of ICE*Bs1*'s regulatory systems and the earlier onset of starvation on DSM agar compared to MSgg agar; cells that are unable to sporulate eventually die.

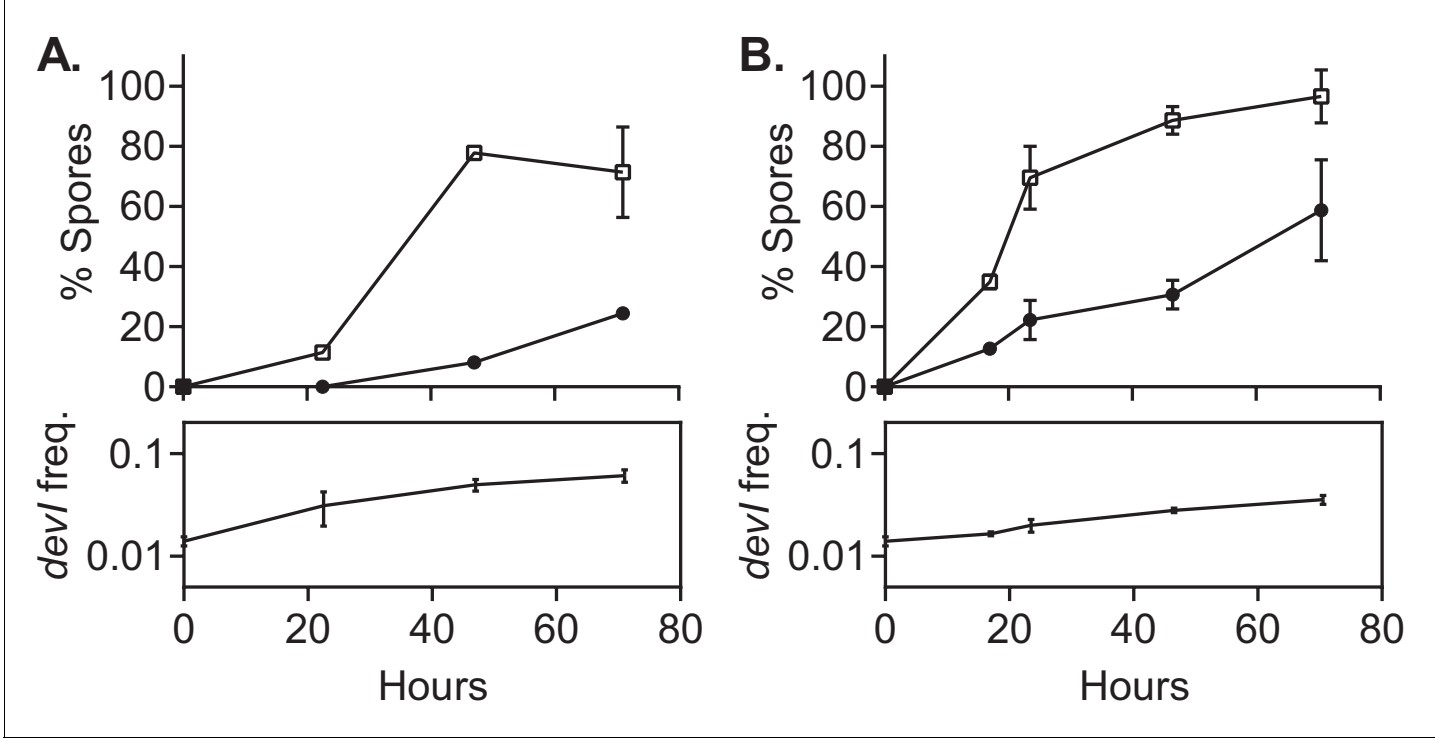

**Figure 6.** *devI* alone is sufficient to inhibit sporulation and provide a selective advantage. ICE*Bs1*-cured cells that constitutively express *devI* (JMJ725, black circles) were mixed at an initial frequency of 0.01 with ICE*Bs1*-cured cells containing an empty expression construct (JMJ727, open squares). The mixture was spotted onto biofilm growth medium (**A**) or sporulation medium (**B**). Biofilms and colonies were harvested at the indicated times to determine the fraction of CFUs derived from heat-resistant spores for both the *devI*-containing and control cells. Boxes below each graph indicate the frequency of *devI*-containing CFUs at each timepoint. Data shown are the average from two populations (biological replicates) per timepoint with error bars indicating the standard deviation.

The online version of this article includes the following source data for figure 6:

**Source data 1.** DevI inhibits sporulation and provides benefit.

### DevI likely targets the developmental transcription factor Spo0A

Results described above demonstrated that *devI* is a robust inhibitor of sporulation. Sporulation is controlled by the transcription factor Spo0A (reviewed in *Hoch, 1993*; *Sonenshein, 2000*}) which both directly and indirectly regulates the expression of many genes needed for development, including biofilm formation (*Hamon and Lazazzera, 2001*). The results described below indicate that DevI most likely targets Spo0A, either directly or indirectly.

#### *devI* inhibits early sporulation gene expression

We found that *devI* inhibits expression of genes normally activated early during sporulation. Sporulation is initiated when Spo0A~P directly stimulates transcription of several genes, including the three sporulation operons, *spoIIA, spoIIE*, and *spoIIG* (*Sonenshein, 2000*). Using *lacZ* fusions to the promoters of each of these operons, we found that P*xis-devI* inhibited activity of each promoter compared to wild-type during sporulation in liquid sporulation medium (*Figure 7A*). This indicates that DevI inhibits the initiation of sporulation, perhaps by affecting the activity or accumulation of Spo0A~P.

#### *devI* inhibits biofilm gene expression

We also found that *devI* inhibits expression of genes needed for extracellular matrix production during biofilm formation. We measured expression of biofilm matrix genes *epsB* and *tasA* (*Hahn et al.,*

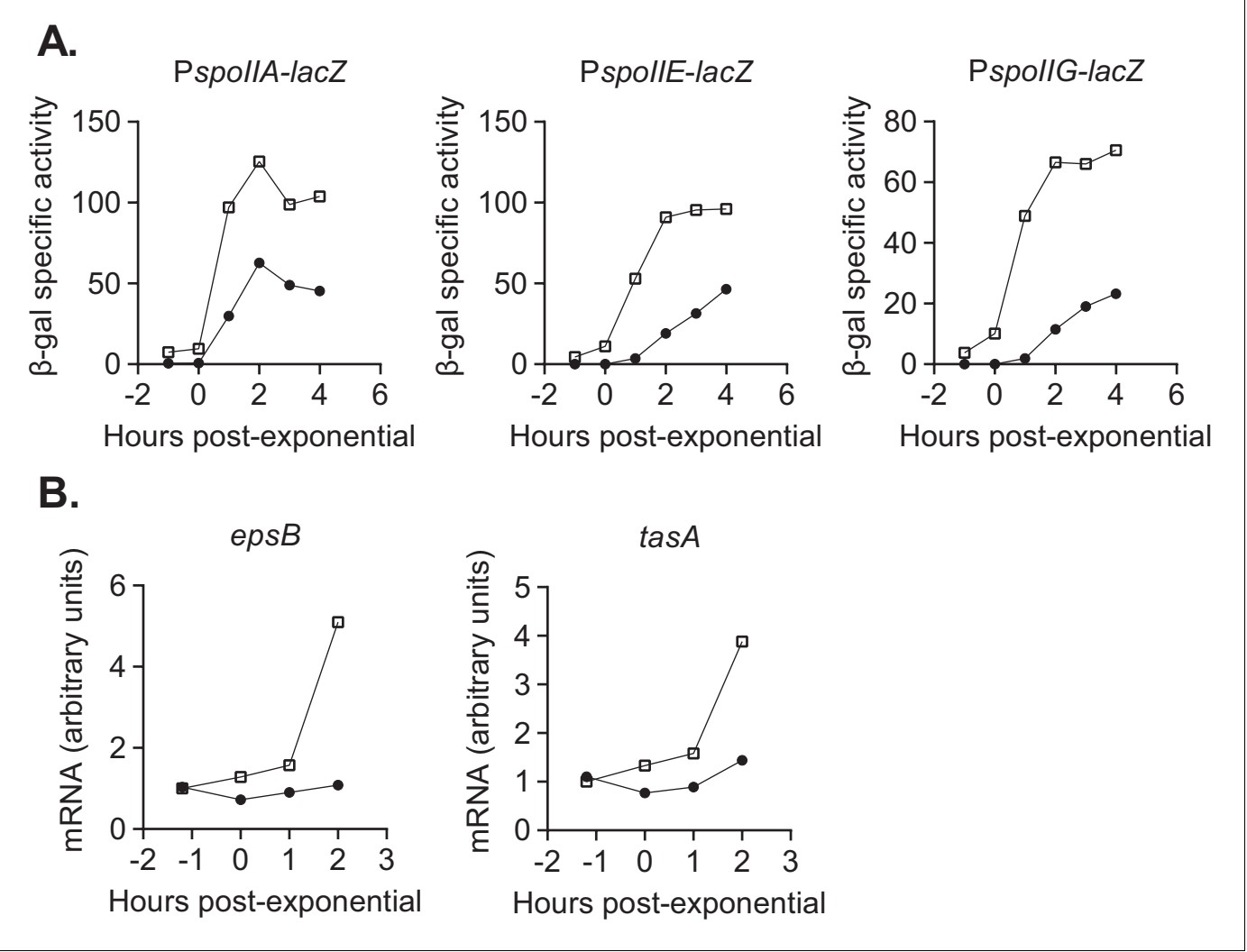

**Figure 7.** *devI* inhibits expression of genes associated with sporulation initiation and biofilm formation. (**A**) ICE*Bs1*-cured cells that constitutively express *devI* (black circles) or contain an empty expression construct (open squares) were grown in liquid sporulation medium. Cells were harvested at the indicated times and β-galactosidase specific activity was measured. Strains: P*spoIIA-lacZ* P*xis-devI* (JMJ732), P*spoIIA-lacZ* WT (JMJ735), P*spoIIE-lacZ* P*xis-devI* (JMJ731), P*spoIIE-lacZ* WT (JMJ734), P*spoIIG-lacZ* P*xis-devI* (JMJ733), P*spoIIG-lacZ* WT (JMJ736). A representative experiment is shown. (**B**) ICE*Bs1*-cured cells that constitutively express *devI* (JMJ725, black circles) and ICE*Bs1*-cured cells containing an empty expression construct (JMJ727, open squares) were grown in liquid biofilm medium, and cells were harvested at the indicated times. cDNA was synthesized using reverse transcriptase and RT-qPCR was used to measure expression of biofilm-associated genes *epsB* and *tasA*. The transcript copy numbers of these genes were measured relative to a housekeeping gene *gyrA*. The data reported are the average of three technical replicates from one experiment. The relative expression levels are normalized to wild-type at $T_{-1}$. A representative experiment is shown.

The online version of this article includes the following source data for figure 7:

**Source data 1.** Sporulation and biofilm gene expression.

1995; *Hamon et al., 2004*; *Bai et al., 1993*; *Kearns et al., 2005*) by RT-qPCR with primers internal to each gene. In early stationary phase in liquid biofilm medium, transcript levels of *epsB* and *tasA* were reduced by about 5-fold and 3-fold, respectively, in the P*xis-devI* strain compared to wild-type (*Figure 7B*). We suspect that inhibition of biofilm matrix genes, in addition to delaying sporulation, is an important mechanism of selection for ICE*Bs1* host cells during growth in a biofilm. This is consistent with the selective advantage of ICE*Bs1* host cells in biofilms without sporulation described

earlier. Inhibition of biofilm and early sporulation genes is consistent with DevI functioning as an inhibitor of Spo0A or its activation by phosphorylation.

### *devI* is conserved among ICEs homologous to ICE*Bs1*

We found that *devI* (*ydcO*) is conserved among *Bacillus* species and in many cases is located within what appear to be ICEs similar to ICE*Bs1*. We used NCBI BLAST to search for homologous protein sequences using both pBLAST (protein database) and tBLASTn (translated nucleotide database). Homologs with 100% sequence coverage and greater than 70% identity to YdcO from *B. subtilis* NCIB3610 were found in dozens of other *B. subtilis* strains and in closely related species including *B. licheniformis, B. atrophaeus,* and *B. amyloliquefaciens.* Excluding *Bacillus* species from the searches to possibly identify more distantly related proteins with known functions produced no hits.

We analyzed the sequence surrounding the *devI* (*ydcO*) homologs identified to determine if there is similarity to ICE*Bs1*. All of the *devI* (*ydcO*) homologs appear to be within mobile element regions similar to ICE*Bs1*, though some are clearly missing genes present in ICE*Bs1*. Although we cannot infer whether any of these regions are functional mobile elements, we suspect that the ability to inhibit host development may be a conserved strategy among ICE*Bs1*-like elements and possibly other ICEs with cargo genes of unknown function.

## Discussion

Our work demonstrates that ICE*Bs1* confers a selective advantage on its host cells by delaying bio-film and spore development, enabling the host to grow more than cells without ICE*Bs1*. When ICE*Bs1*-containing cells are the minority in a mixed population, ICE*Bs1* genes are induced. One of these genes, *devI*, is necessary and sufficient to inhibit biofilm- and sporulation-associated gene expression, likely by inhibiting the key developmental regulator Spo0A, either directly or indirectly. Together with previous findings we conclude that ICE*Bs1* encodes at least three distinct strategies to benefit its host cells. (1) Inhibition of development (described here) provides a growth advantage in biofilms and during sporulation. (2) Exclusion, mediated by *yddJ*, blocks the conjugation machin-ery and protects the host cell from lethal excessive transfer (*Avello et al., 2019*). (3) An abortive infection mechanism, mediated by *spbK* (*yddK*) protects populations of ICE*Bs1* host cells from pre-dation by the lysogenic phage SPβ (*Johnson et al., 2020*). We propose that all three strategies pro-vide a competitive advantage for ICE*Bs1* and its host cells in different conditions.

Expression of *devI* reduces biofilm matrix expression and delays the initiation of sporulation. Pro-duction of the biofilm matrix is a public good, benefiting the whole community (*Dragoš et al., 2018*). Avoidance of matrix production can therefore be considered an exploitative behavior. Exploi-tation can be detrimental to the population as a whole (*Smith and Schuster, 2019*), but we did not observe any negative effects on populations under conditions where ICE*Bs1* host cells had an advan-tage. This is in agreement with the facultative nature of ICE*Bs1* cheating (*Even-Tov et al., 2016*; *Pollak et al., 2016*). Quorum-sensing by *rapI-phrI* ensures that ICE*Bs1* cheats only as a minority, where its impact on total public goods levels is negligible. Interestingly, the pBS32 plasmid utilizes direct regulation of biofilm formation by a Rap receptor to its benefit (*Omer Bendori et al., 2015*; *Pollak et al., 2015*), while in ICE*Bs1* this regulation was moved from the Rap receptor to one of its regulated genes.

The fitness consequences of sporulation inhibition are complicated (*Mutlu et al., 2018*). Delaying sporulation too long would result in a loss of viability of the starved cells. Inhibition of sporulation by ICE*Bs1* appears to be transient; ICE*Bs1* host cells eventually sporulate and do not lose significant viability as a consequence of delaying sporulation. Regulation of *devI* expression by the cell-cell sig-naling genes *rapI-phrI* is likely critical for transient developmental inhibition. Because commitment to sporulation is irreversible, sporulating too early is detrimental if nutrient deprivation is short-lived. *B. subtilis* cells with activated Spo0A that have not yet committed to sporulate also delay commitment to sporulation by killing sibling cells to liberate nutrients ('cannibalism') (*González-Pastor et al., 2003*). Cannibalism is regulated by Spo0A and the subpopulation of cannibal cells (those doing the killing) overlaps with those producing the biofilm matrix (*López et al., 2009*). Because of this over-lap, it seems unlikely that *devI* delays sporulation by stimulating cannibalism.

## ICE*Bs1* stability during sporulation

We hypothesize that in addition to providing a fitness advantage to its host cell, delaying sporulation may also improve stability of ICE*Bs1* in the host during development. Sporulation involves the formation of an asymmetric division septum generating the larger mother cell and the smaller forespore (*Errington, 2001*; *Higgins and Dworkin, 2012*). Sporulation is induced when cells are at a high population density and running out of nutrients, conditions that also activate ICE*Bs1* (*Auchtung et al., 2005*; *Grossman and Losick, 1988*). The plasmid form of ICE*Bs1* that is generated after excision from the chromosome is not known to have a mechanism for active partitioning and is more likely to remain in the larger mother cell if the cells do enter the sporulation pathway and divide asymmetrically. Therefore, the ability of ICE*Bs1* to delay the initiation of sporulation under conditions where the element is activated could help prevent loss of the element and maintain ICE*Bs1* in host cells.

Mobile genetic elements employ various strategies to promote their maintenance during sporulation. Rates of curing during sporulation for various plasmids in *Bacillus* species vary widely and do not necessarily correlate with their stability during normal cell division (*Tokuda et al., 1993*; *Turgeon et al., 2008*). Mechanisms encoded by plasmids to promote their stability during growth and sporulation include the production of dynamic cytoskeletal filaments (*Becker et al., 2006*) and post-segregational killing of plasmid-cured pre-spores with toxin-antitoxin systems (*Short et al., 2015*). Interestingly, even lytic phage genomes can be incorporated into spores (first described in the 1960 s) by co-opting the host's chromosomal partitioning system (*Meijer et al., 2005*).

## Diversity of cargo genes and associated phenotypes

Mobile genetic elements, especially ICEs, are widespread in bacteria (*Frost et al., 2005*; *Guglielmini et al., 2011*). Many known mobile genetic elements encode cargo genes that confer easily recognizable phenotypes, notably antibiotic resistance. Other cargo genes provide less obvious phenotypes but still fundamentally alter the physiology of the host cell. A large (500 kb) ICE was discovered in *Mesorhizobium loti* because its horizontal transfer conferred the ability to form nitrogen-fixing symbiotic rood nodules on *Lotus* plant species (*Sullivan and Ronson, 1998*). In many pathogens, cargo genes in mobile elements are largely responsible for virulence. For example, *Vibrio cholerae* is capable of a pathogenic lifestyle in human hosts due to the toxin-encoding phage CTXΦ (*Waldor and Mekalanos, 1996*). In the sporulating pathogen *Bacillus anthracis*, mobile genetic elements regulate both virulence and host development. Two plasmids, pXO1 and pXO2, provide the genes for toxin synthesis and production of a protective capsule, respectively (*Green et al., 1985*; *Mikesell et al., 1983*). pXO1 also contains a regulatory gene, *atxA*, that regulates virulence factor production and inhibits host cell sporulation (*Dale et al., 2018*). Co-regulation of virulence factors and sporulation is likely important during infection, as *B. anthracis* spores are thought to be more susceptible than vegetative cells to eradication by the immune system (*Mock and Fouet, 2001*).

Mobile elements are also known to alter the host's interaction with other horizontally acquired DNA, which has implications for the fitness and evolvability of the host. For example, the plasmid pBS32 in *B. subtilis* encodes an inhibitor of the host's DNA uptake machinery, blocking natural transformation (*Konkol et al., 2013*). Interestingly, genes with roles in defense against foreign DNA, CRISPR-Cas systems, are also identified within mobile elements (*Faure et al., 2019*; *McDonald et al., 2019*; *Millen et al., 2012*). Competition between mobile elements not only shapes the repertoire of cargo genes in a given cell, but it may also protect the host from harmful elements.

Many mobile genetic elements have been identified bioinformatically from genome sequences or discovered by means other than the phenotypes they provide (*Bi et al., 2012*; *Guglielmini et al., 2011*; *Johnson and Grossman, 2015*). Many elements lack obvious cargo genes, or at least lack cargo genes that have recognizable functions (*Cury et al., 2017*). We suspect that many elements with uncharacterized cargo genes provide important traits to their hosts beyond the scope of the phenotypes currently attributed to mobile elements. Although mobile genetic elements can have remarkably broad host ranges, such as the Tn*916*-Tn*1545* group of ICEs (*Clewell et al., 1995*; *Roberts and Mullany, 2009*) and the IncP-1 group of plasmids (*Popowska and Krawczyk-Balska, 2013*), cargo genes and their associated functions could be highly specific to certain hosts.

Characterization of unknown cargo genes is likely to expand the diversity of traits currently attributed to mobile genetic elements. We speculate that many of these genes modulate normal host

functions rather than provide entirely new phenotypes. Understanding cargo gene function is critical for understanding interactions between and co-evolution of mobile elements and their hosts.

# Materials and methods

## Key resources table

| Reagent type (species) or resource | Designation | Source or reference | Identifiers | Additional information |
|---|---|---|---|---|
| Strain, strain background (*Bacillus subtilis* NCIB3610) | DS2569 | *Konkol et al., 2013*. PMID:23836866 | | NCIB3610 cured of pBS32 plasmid. Gift of Daniel Kearns to Avigdor Eldar. |
| Strain, strain background (*Bacillus subtilis* DS2569) | JMJ550 | This paper | | ICE*Bs1*[0] *lacA*::{Ppen-*mApple2 kan*} |
| Strain, strain background (*Bacillus subtilis* DS2569) | JMJ574 | This paper | | ICE*Bs1*[0] *lacA*::{Pveg-*mTagBFP mls*} |
| Strain, strain background (*Bacillus subtilis* DS2569) | JMJ576 | This paper | | ICE*Bs1*[0] *bcaP*::{P*rapI-rapIphrI kan*} *lacA*::{Pveg-*mTagBFP mls*} |
| Strain, strain background (*Bacillus subtilis* DS2569) | JMJ592 | This paper | | ICE*Bs1* *yddJ-cat-yddK lacA*::{Pveg-*mTagBFP mls*} |
| Strain, strain background (*Bacillus subtilis* DS2569) | JMJ593 | This paper | | ICE*Bs1* *conEK476E yddJ-cat-yddK lacA*::{Pveg-*mTagBFP mls*} |
| Strain, strain background (*Bacillus subtilis* DS2569) | JMJ646 | This paper | | ICE*Bs1* *oriT\* attR*::*tet lacA*::{Pveg-*mTagBFP mls*} |
| Strain, strain background (*Bacillus subtilis* DS2569) | JMJ662 | This paper | | ICE*Bs1* Δ*Pxis oriT\* attR*::*tet lacA*::{Pveg-*mTagBFP mls*} |
| Strain, strain background (*Bacillus subtilis* DS2569) | JMJ686 | This paper | | ICE*Bs1* *oriT\* ΔrapIphrI attR*::*tet lacA*::{Pveg-*mTagBFP mls*} |
| Strain, strain background (*Bacillus subtilis* DS2569) | JMJ688 | This paper | | ICE*Bs1* *oriT\* ΔsncO attR*::*tet lacA*::{Pveg-*mTagBFP mls*} |
| Strain, strain background (*Bacillus subtilis* DS2569) | JMJ703 | This paper | | ICE*Bs1* *oriT\* ΔdevI attR*::*tet lacA*::{Pveg-*mTagBFP mls*} |
| Strain, strain background (*Bacillus subtilis* DS2569) | JMJ704 | This paper | | ICE*Bs1* *oriT\* ΔydzL attR*::*tet lacA*::{Pveg-*mTagBFP mls*} |
| Strain, strain background (*Bacillus subtilis* DS2569) | JMJ714 | This paper | | ICE*Bs1*[0] *lacA*::*spec bcaP*::*kan* |
| Strain, strain background (*Bacillus subtilis* DS2569) | JMJ725 | This paper | | ICE*Bs1*[0] *lacA*::{Pxis-*devI mls*} |
| Strain, strain background (*Bacillus subtilis* DS2569) | JMJ727 | This paper | | ICE*Bs1*[0] *lacA*::{Pxis-*empty mls*} |
| Strain, strain background (*Bacillus subtilis* DS2569) | JMJ731 | This paper | | ICE*Bs1*[0] *lacA*::{Pxis-*devI mls*} *amyE*::{PspoIIE-*lacZ cat*} |
| Strain, strain background (*Bacillus subtilis* DS2569) | JMJ732 | This paper | | ICE*Bs1*[0] *lacA*::{Pxis-*devI mls*} *amyE*::{PspoIIA-*lacZ cat*} |
| Strain, strain background (*Bacillus subtilis* DS2569) | JMJ733 | This paper | | ICE*Bs1*[0] *lacA*::{Pxis-*devI mls*} *amyE*::{PspoIIG-*lacZ cat*} |
| Strain, strain background (*Bacillus subtilis* DS2569) | JMJ734 | This paper | | ICE*Bs1*[0] *lacA*::{Pxis-*empty mls*} *amyE*::{PspoIIE-*lacZ cat*} |
| Strain, strain background (*Bacillus subtilis* DS2569) | JMJ735 | This paper | | ICE*Bs1*[0] *lacA*::{Pxis-*empty mls*} *amyE*::{PspoIIA-*lacZ cat*} |
| Strain, strain background (*Bacillus subtilis* DS2569) | JMJ736 | This paper | | ICE*Bs1*[0] *lacA*::{Pxis-*empty mls*} *amyE*::{PspoIIG-*lacZ cat*} |
| Strain, strain background (*Bacillus subtilis* DS2569) | JMJ785 | This paper | | ICE*Bs1* *oriT\* ΔrapIphrI attR*::*tet bcaP*::{P*rapI-rapIphrI kan*} *lacA*::{Pveg-*mTagBFP mls*} |

*Continued on next page*

*Continued*

| Reagent type (species) or resource | Designation | Source or reference | Identifiers | Additional information |
|---|---|---|---|---|
| Strain, strain background (*Bacillus subtilis* DS2569) | JMJ786 | This paper | | ICE*Bs1*[0] *spo0A∆Ps lacA*::{P*pen*-*mApple2 kan*} |
| Strain, strain background (*Bacillus subtilis* DS2569) | JMJ788 | This paper | | ICE*Bs1 conEK476E yddJ-cat-yddK spo0A∆Ps lacA*::{P*veg*-*mTagBFP mls*} |
| Strain, strain background (*Escherichia coli* MC1061) | AG1111 | *Ireton et al., 1993* PMID:8436298 | | *E. coli* strain for cloning and maintaining plasmids. MC1061 with F′ *proAB*+ *lacI*q *lacZM15* Tn*10*. |
| Recombinant DNA reagent | pJMJ196 (plasmid) | This paper | | For generating oriT* *nick*; derived from pCAL1422. |
| Recombinant DNA reagent | pJMJ430 (plasmid) | This paper | | For generating unmarked *rapI-phrI* deletion; derived from pCAL1422. |
| Recombinant DNA reagent | pJMJ199 (plasmid) | This paper | | For generating unmarked P*xis* deletion; derived from pCAL1422. |
| Recombinant DNA reagent | pELS5 (plasmid) | Other | | For generating unmarked *ydzL* deletion; derived from pCAL1422. From Grossman lab collection. |
| Recombinant DNA reagent | pELS1 (plasmid) | Other | | For generating unmarked *devI* deletion; derived from pCAL1422. From Grossman lab collection. |
| Recombinant DNA reagent | pELC815 (plasmid) | Other | | For generating unmarked *sncO* deletion; derived from pCAL1422. From Grossman lab collection. |
| Recombinant DNA reagent | pJT245 (plasmid) | Other | | Source of oriT* *nicK* allele. From Grossman lab collection. |
| Recombinant DNA reagent | pCAL1422 (plasmid) | *Thomas et al., 2013*. PMID:23326247 | | For generating markerless deletions/mutations. |
| Recombinant DNA reagent | pMMH253 (plasmid) | Other | | Vector for integration of constructs at *bcaP*. From Grossman lab collection. |
| Recombinant DNA reagent | pJMJ354 (plasmid) | This paper | | Native *rapI-phrI* expression construct for integration at *bcaP*; derived from pMMH253 |
| Sequence-based reagent | oJJ363 | Sigma-Aldrich | qPCR primer | CGGAACAATATCGCACCATTC |
| Sequence-based reagent | oJJ364 | Sigma-Aldrich | qPCR primer | CGCTGCACTGAACGATTTAC |
| Sequence-based reagent | oJJ367 | Sigma-Aldrich | qPCR primer | GGATCACTTGCGATCAAAGAAG |
| Sequence-based reagent | oJJ368 | Sigma-Aldrich | qPCR primer | CTTCAAACTGGCTGAGGAAATC |
| Sequence-based reagent | oMEA128 | Sigma-Aldrich | qPCR primer | TGGAGCATTACCTTGACCATC |
| Sequence-based reagent | oMEA129 | Sigma-Aldrich | qPCR primer | AGCTCTCGCTTCTGCTTTAC |
| Commercial assay or kit | RNeasy PLUS | Qiagen | Cat No. 74136 | |
| Commercial assay or kit | iScript Reverse Transcription Supermix | Bio-Rad | Cat No. 1708840 | |
| Commercial assay or kit | SsoAdvanced SYBR master mix | Bio-Rad | Cat No. 1725270 | |

## Media and growth conditions

Prior to competition experiments, cells were grown as light lawns for approximately 20 hr at room temperature on 1.5% agar plates containing 1% w/v glucose, 0.1% w/v monopotassium glutamate,

and 1x Spizizen's salts (2 g/l $(NH_4)SO_4$, 14 g/l $K_2HPO_4$, 6 g/l $KH_2PO_4$, 1 g/l $Na_3$citrate-$2H_2O$, and 0.2 g/l $MgSO_4$-$7H_2O$) (*Harwood and Cutting, 1990*). Cells were resuspended from light lawns and grown at 37°C with shaking in S7$_{50}$ defined minimal medium (*Jaacks et al., 1989*) with 1% w/v glucose and 0.1% w/v monopotassium glutamate. Biofilms were grown at 30°C on MSgg agar plates (as defined in *Branda et al., 2001* with the exception of tryptophan and phenylalanine, which we did not include). The sporulation medium used was DSM (in liquid form or as plates solidified with 1.5% agar) (*Harwood and Cutting, 1990*). MSgg agar and DSM agar plates were dried for 20–24 hr at 37°C prior to use. Antibiotics were used at the following concentrations for selection on LB agar plates: chloramphenicol (5 µg/ml), kanamycin (5 µg/ml), spectinomycin (100 µg/ml), tetracycline (12.5 µg/ml), and a combination of erythromycin (0.5 µg/ml) and lincomycin (12.5 µg/ml) to select for macrolide-lincosamide-streptogramin (MLS) resistance.

## Strains and alleles

The *B. subtilis* strains used are listed in *Table 2*. The strain background used in all experiments was a derivative of the undomesticated strain NCIB3610 lacking its endogenous plasmid pBS32 (NCIB3610 plasmid-free). ICE*Bs1*$^0$ indicates the strain is cured of ICE*Bs1*. Standard techniques were used for cloning and strain construction (*Harwood and Cutting, 1990*). Some alleles were previously described and are summarized below. Variants of ICE*Bs1* that were blocked for transfer contained the *conEK476E* mutation derived from MMB1118 (*Berkmen et al., 2010*). The *spo0AΔPs* allele was derived from AG1242 (*Siranosian and Grossman, 1994*). The *amyE*::P*spoIIA-lacZ cat* allele was derived from KI938 (*Chung et al., 1994*). Essentially identical alleles with P*spoIIE* and P*spoIIG* were also used.

**Table 2.** *B. subtilis* strains used[*].

| Strain | Relevant genotype |
|---|---|
| JMJ550 | ICE*Bs1*$^0$ *lacA*::{Ppen-*mApple2 kan*} |
| JMJ574 | ICE*Bs1*$^0$ *lacA*::{Pveg-*mTagBFP mls*} |
| JMJ576 | ICE*Bs1*$^0$ *bcaP*::{PrapI-*rapIphrI kan*} *lacA*::{Pveg-*mTagBFP mls*} |
| JMJ592 | ICE*Bs1 yddJ-cat-yddK lacA*::{Pveg-*mTagBFP mls*} |
| JMJ593 | ICE*Bs1 conEK476E yddJ-cat-yddK lacA*::{Pveg-*mTagBFP mls*} |
| JMJ646 | ICE*Bs1 oriT* attR::tet lacA*::{Pveg-*mTagBFP mls*} |
| JMJ662 | ICE*Bs1 ΔPxis oriT* attR::tet lacA*::{Pveg-*mTagBFP mls*} |
| JMJ686 | ICE*Bs1 oriT* ΔrapIphrI attR::tet lacA*::{Pveg-*mTagBFP mls*} |
| JMJ688 | ICE*Bs1 oriT* ΔsncO attR::tet lacA*::{Pveg-*mTagBFP mls*} |
| JMJ703 | ICE*Bs1 oriT* ΔdevI attR::tet lacA*::{Pveg-*mTagBFP mls*} |
| JMJ704 | ICE*Bs1 oriT* ΔydzL attR::tet lacA*::{Pveg-*mTagBFP mls*} |
| JMJ714 | ICE*Bs1*$^0$ *lacA*::spec *bcaP*::kan |
| JMJ725 | ICE*Bs1*$^0$ *lacA*::{Pxis-*devI mls*} |
| JMJ727 | ICE*Bs1*$^0$ *lacA*::{Pxis-empty *mls*} |
| JMJ731 | ICE*Bs1*$^0$ *lacA*::{Pxis-*devI mls*} *amyE*::{P*spoIIE-lacZ cat*} |
| JMJ732 | ICE*Bs1*$^0$ *lacA*::{Pxis-*devI mls*} *amyE*::{P*spoIIA-lacZ cat*} |
| JMJ733 | ICE*Bs1*$^0$ *lacA*::{Pxis-*devI mls*} *amyE*::{P*spoIIG-lacZ cat*} |
| JMJ734 | ICE*Bs1*$^0$ *lacA*::{Pxis-empty *mls*} *amyE*::{P*spoIIE-lacZ cat*} |
| JMJ735 | ICE*Bs1*$^0$ *lacA*::{Pxis-empty *mls*} *amyE*::{P*spoIIA-lacZ cat*} |
| JMJ736 | ICE*Bs1*$^0$ *lacA*::{Pxis-empty mls} *amyE*::{P*spoIIG-lacZ cat*} |
| JMJ785 | ICE*Bs1 oriT* ΔrapIphrI attR::tet bcaP*::{PrapI-*rapIphrI kan*} *lacA*::{Pveg-*mTagBFP mls*} |
| JMJ786 | ICE*Bs1*$^0$ *spo0AΔPs lacA*::{Ppen-*mApple2 kan*} |
| JMJ788 | ICE*Bs1 conEK476E yddJ-cat-yddK spo0AΔPs lacA*::{Pveg-*mTagBFP mls*} |

[1]All strains derived from NCIB3610 plasmid-free.

## Construction of selective markers for mating and competition experiments

ICE*Bs1* was marked with the *cat* gene (conferring chloramphenicol resistance) between the divergently transcribed genes *yddJ* and *spbK* (*yddK*). Markers used to select ICE*Bs1*-containing and ICE*Bs1*⁰ cells were all integrated at *lacA* and contained *spec* (spectinomycin resistance), *mls* (macrolide-lincosamide-streptogramin resistance), or *kan* (kanamycin resistance). The *mls* and *kan* constructs also contained constitutively expressed fluorescent proteins BFP and RFP, respectively. The plating efficiency of all markers was verified, and control competitions (described below) were performed to measure marker-associated fitness effects.

## Construction of ICE*Bs1* mutants

Fitness measurements of ICE*Bs1* mutants were performed in a version of ICE*Bs1* unable to excise and replicate (locked-in ICE*Bs1*). All mutants were isogenic to JMJ646 (ICE*Bs1* oriT* Δ*attR::tet*), which is unable to excise due to the *attR::tet* deletion (*Lee and Grossman, 2007*). The origin of transfer was mutated (oriT*) to prevent ICE*Bs1* replication while integrated, which is detrimental (*Lee and Grossman, 2007*; *Menard and Grossman, 2013*). The markerless oriT* mutation was constructed by cloning *nicK*(oriT*) from pJT245 and ~1 kb of upstream sequence into pCAL1422 (a plasmid that contains *E. coli lacZ*) by isothermal assembly (*Gibson et al., 2009*), essentially as previously described (*Thomas et al., 2013*). The resulting plasmid, pJMJ196, was integrated into the chromosome by single-crossover recombination. Transformants were screened for loss of *lacZ*, indicating loss of the integrated plasmid, and PCR was used to identify a clone containing the oriT* allele. Markerless deletions of ICE*Bs1* sequence were also generated using pCAL1422-derived plasmids. The *rapI-phrI* deletion was generated using pJMJ430 and removes the *rapI* and *phrI* ORFs. The P*xis* deletion was generated using pJMJ199 and removes sequence from 149 bp to 27 bp upstream of the *xis* ORF. This removes the presumed −35 and −10 of the promoter but does not remove the known regulatory sites at the neighboring *immR* promoter (*Auchtung et al., 2007*). The *ydzL* deletion was generated using pELS5 and fuses the first four and last two codons of *ydzL*. The *devI* deletion was generated using pELS1 and fuses the first four and last two codons of *devI*. The *sncO* deletion was generated using pELC815 and removes the *sncO* gene and 44 bp of upstream sequence.

## Construction of ectopic *rapI-phrI* construct

The *rapI-phrI* ORFs plus the promoter region (352 bp upstream of *rapI*) and 112 bp of downstream sequence were cloned into pMMH253 (vector for integration at *bcaP*). The resulting plasmid (pJMJ354) was linearized and introduced to *B. subtilis* by transformation and selection for kanamycin resistance. The corresponding empty control construct was generated by transforming linearized pMMH253.

## Construction of *lacA::Pxis-devI*

We expressed *devI* from the P*xis* promoter by cloning existing elements from a P*xis* gene expression construct marked with *mls* at *lacA* and inserting the *devI* ORF by isothermal assembly. The resulting product was introduced to the chromosome by transformation and selection for MLS resistance. The P*xis*-empty control strain contains an identical construct lacking an ORF fused to P*xis*.

## Biofilm mating experiments

Cultures were started from resuspended light lawns (described above) diluted to an initial OD600 of 0.05 in S7₅₀ minimal medium. Cultures were grown to mid-exponential phase (OD600 ~0.5) at 37°C with shaking. Cells were pelleted, resuspended in 1x Spizizen's salts, and diluted to an OD600 of 0.01. Donor and recipient strains were mixed at the indicated frequencies and 50 µl of the mixture was spotted onto the center of MSgg agar plates. Spots were allowed to dry at 30°C before flipping the plates. Plates were incubated at 30°C for 4 days. At the time of inoculation, the strain mixes were diluted and plated in duplicate on LB agar plates containing the appropriate antibiotics to determine the initial CFU/ml of the donor and recipient strains. After 4 days, the mature biofilms were scraped from the agar surface with sterile wooden sticks and resuspended in 5 ml 1x Spizizen's salts, followed by mild sonication to disperse the cells. Cells were diluted and selectively plated to determine the final CFU/ml of transconjugants.

## Competition experiments

Cells were prepared for competition experiments as described above for biofilm mating experiments. Strain mixtures at the indicated frequencies were spotted onto MSgg agar plates for biofilm competitions and DSM agar plates for sporulation competitions. Plates were incubated at 30°C for 4 days unless otherwise indicated. Biofilms/colonies were collected, disrupted, and plated as described above. For time-course competitions, two replicate biofilms/colonies were collected at each of the indicated times. Sporulation frequency was determined by selective plating before and after a heat treatment at 85°C for 20 min to enumerate total CFUs and CFUs derived from heat-resistant spores. Relative fitness of ICE*Bs1*-containing cells over ICE*Bs1*$^0$ cells was determined as $(p_f/(1-p_f))/(p_i/(1-p_i))$, where $p_f$, $p_i$ are the frequencies of ICE*Bs1*-containing cells in the final and initial populations, respectively. Control competitions between ICE*Bs1*-cured cells were performed to determine the fitness associated with the *lacA*::{Pveg-*mTagBFP mls*} marker (JMJ574) used in ICE*Bs1*-containing cells relative to the *lacA*::{Ppen-*mApple2 kan*} marker used in ICE*Bs1*-null cells (JMJ550). When the *mls*-marked cells were started at a frequency of 0.01, their relative fitness was 0.7 ± 0.09 (average and standard deviation from three independent experiments and a total of 9 biofilms).

## Gene expression assays

Expression of sporulation genes was measured in cultures grown from single colonies in liquid DSM at 37° with shaking. Cells were harvested at the indicated timepoints. For β-galactosidase assays, growth was stopped by the addition of toluene (~1.5% final concentration). β-galactosidase specific activity ([$\Delta A_{420}$ per minute per ml of culture per $OD_{600}$] x 1000) was measured as described (*Miller, 1972*) after pelleting cell debris.

Biofilm gene expression was measured in cultures grown from single colonies in liquid MSgg at 37° with shaking. For RT-qPCR assays, cells were harvested directly into ice-cold methanol (1:1 methanol to culture volume) and pelleted. RNA was isolated using Qiagen RNeasy PLUS kit with 10 mg/ml lysozyme. iScript Supermix (Bio-Rad) was used for reverse transcriptase reactions to generate cDNA. Control reactions without reverse transcriptase were performed to assess the amount of DNA present in the RNA samples. RNA was degraded by adding 75% vol of 0.1 M NaOH and incubating at 70°C for 10 min, followed by neutralization with an equal volume of 0.1 M HCl. qPCR was done using SSoAdvanced SYBR master mix and CFX96 Touch Real-Time PCR system (Bio-Rad). Primers used to measure *epsB* were oJJ363 (5'-CGGAACAATATCGCACCATTC-3') and oJJ364 (5'-CGCTGCACTGAACGATTTAC-3'). Primers used to quantify *tasA* were oJJ367 (5'-GGATCACTTGCGATCAAAGAAG-3') and oJJ368 (5'-CTTCAAACTGGCTGAGGAAATC-3'). Primers used to measure the control locus *gyrA* were oMEA128 (5'-TGGAGCATTACCTTGACCATC-3') and oMEA129 (5'-AGCTCTCGCTTCTGCTTTAC-3'). The relative transcript copy numbers (as indicated by the Cp values measured by qPCR) of *epsB* and *tasA* were normalized to *gyrA* after subtracting the signal from control reactions without reverse transcriptase.

## Acknowledgements

We thank Mary Anderson and Janet Smith for helpful comments on the manuscript. Research reported here is based upon work supported, in part, by the National Institute of General Medical Sciences of the National Institutes of Health under award number R01GM050895 and R35 GM122538 to ADG. Any opinions, findings, and conclusions or recommendations expressed in this report are those of the authors and do not necessarily reflect the views of the National Institutes of Health. Funding for travel was provided by the MIT International Science and Technology Initiatives (MISTI) MIT-Israel Seed Fund.

## Additional information

### Funding

| Funder | Grant reference number | Author |
| --- | --- | --- |
| National Institute of General Medical Sciences | R01 GM050895 | Alan D Grossman |

| National Institute of General Medical Sciences | R35 GM122538 | Alan D Grossman |

The funders had no role in study design, data collection and interpretation, or the decision to submit the work for publication.

## Author contributions

Joshua M Jones, Conceptualization, Formal analysis, Validation, Investigation, Visualization, Methodology, Writing - original draft, Writing - review and editing; Ilana Grinberg, Investigation, Methodology; Avigdor Eldar, Conceptualization, Formal analysis, Supervision, Funding acquisition, Writing - review and editing; Alan D Grossman, Conceptualization, Supervision, Funding acquisition, Visualization, Project administration, Writing - review and editing

## Author ORCIDs

Joshua M Jones http://orcid.org/0000-0002-3327-8899
Avigdor Eldar https://orcid.org/0000-0001-8485-9370
Alan D Grossman https://orcid.org/0000-0002-8235-7227

## Decision letter and Author response

Decision letter https://doi.org/10.7554/eLife.65924.sa1
Author response https://doi.org/10.7554/eLife.65924.sa2

# Additional files

## Supplementary files

• Source data 1. Neutral fitness due to marker effects. Counts of two ICEBs1-cured strains each bearing an antibiotic resistance marker (used to distinguish strains) at the start and end of competitions.

## Data availability

All data generated or analysed during this study are included in the manuscript and supporting files. Source data files have been provided for all figures.

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
