## [Decision Letter]

**Acceptance summary:**

This is an exceptionally rigorous paper that sets an important precedent for how mobile genetic elements can influence host biology. Significantly, it points to the existence of an uncharacterized universe of genes on mobile elements that modulate the lives of their host cells in fascinating and ecologically important ways. We look forward to learning what this group and others discover about these ancient relationships in future studies.

**Decision letter after peer review:**

Thank you for submitting your article "A mobile genetic element increases bacterial host fitness by manipulating development" for consideration by *eLife*. Your article has been reviewed by three peer reviewers, and the evaluation has been overseen by a Reviewing Editor and Gisela Storz as the Senior Editor. The following individuals involved in review of your submission have agreed to reveal their identity: David Dubnau (Reviewer #3); Gary Dunny (Reviewer #4).

Summary:

All the reviewers were in agreement that this is an exceptionally rigorous paper that sets an important precedent for how mobile genetic elements can influence hosts biology. The only requested revisions are the inclusion of a model figure to facilitate comprehension by a general audience and a few changes to one figure and the Discussion. Please feel free to address any other suggestions from the reviewers as you see fit, but do not feel obligated to do any additional experiments.

Essential Revisions:

1) The addition of a model figure pointing out the key players in ICE function and regulation to the Introduction was deemed valuable by all three reviewers.

2) The fitness values for ICE + cells in biofilms without sporulation are reported in text only (subsection “ICE*Bs1* confers a selective advantage in biofilms without sporulation”). Incorporating these data into Figure 1 using the same format would be beneficial.

3) There was consensus that conclusion would benefit from a brief discussion of how *devI* might fit into other models of development in *B. subtilis* (e.g. cannibalism).

Reviewer #2 (Recommendations for the authors):

I truly enjoyed reading the paper and think it's well-executed.

Suggestions for DevI mechanism characterization: Is it feasible to perform a Spo0A immunoblot to assess Spo0A protein levels and perhaps phosphorylation status? KinA overexpression stimulates early sporulation (Fujita Genes Dev 2005), is DevI epistatic to KinA overexpression (and therefore acts downstream)? Perhaps it is worth doing bacterial two-hybrid assays to detect interactions between DevI and Spo0A or other phosphorelay components?

The fitness values for ICE*Bs1*+ cells in biofilms without sporulation are reported in text only (subsection “ICE*Bs1* confers a selective advantage in biofilms without sporulation”). I recommend they be incorporated into Figure 1 using the same format for the data already present.

Reviewer #4 (Recommendations for the authors):

I have only a few questions and suggestions for the authors to consider.

– Since *eLife* is targeted to a broad audience, I think that the detailed discussion of ICE biology and other aspects of the lengthy Introduction would be more digestible if accompanied by a model figure pointing out the key players in ICE function and regulation.

– In the data reported in Figure 4, I was a little surprised that *Δ-sncO* was less competitive than wild type – I would expect that the convergent transcription of this locus might antagonize transcripts from P*xis*, reducing *dev1* expression, so I would expect a stronger effect of Dev1 in the deletion of *sncO* relative to wild type – (maybe the 9.3 vs. 13.8 numbers are really not different, but the authors should comment)

– Over-expression of Dev1 from a constitutive promoter clearly had an effect of increasing fitness of the host cell; given the extensive tools available in B subtilis, it would be interesting to use a promoter whose expression can be varied by inducer levels to get an idea of the amount of Dev 1 protein required to inhibit Spo0A – they could start with the lac reporter and then proceed to sporulation/biofilm assays – maybe this is for the next paper, but it is an interesting question related to mechanism.

– Finally, the Losick lab published work several years ago on B subtilis cannibalism, where a subfraction of developing cells kill siblings, allowing the killers to obtain additional nutrients and delay sporulation (if I recall correctly). Does the Dev1 system fit into the biology of cannibalism? Authors might comment in the Discussion.

---

## [Author Response]

Essential Revisions:1) The addition of a model figure pointing out the key players in ICE function and regulation to the Introduction was deemed valuable by all three reviewers.

We added a new figure, now Figure 1, to the Introduction. Subsequent figures renumbered accordingly.

2) The fitness values for ICEBs1+ cells in biofilms without sporulation are reported in text only (subsection “ICEBs1 confers a selective advantage in biofilms without sporulation”). Incorporating these data into Figure 1 using the same format would be beneficial.

These have been incorporated into what is now Figure 2E and F.

3) There was consensus that conclusion would benefit from a brief discussion of how devI might fit into other models of development in *B. subtilis* (e.g. cannibalism).

A brief discussion about cannibalism has been added to the Discussion. We also added a couple of references.

Reviewer #2 (Recommendations for the authors):I truly enjoyed reading the paper and think it's well-executed.Suggestions for DevI mechanism characterization: Is it feasible to perform a Spo0A immunoblot to assess Spo0A protein levels and perhaps phosphorylation status? KinA overexpression stimulates early sporulation (Fujita Genes Dev 2005), is DevI epistatic to KinA overexpression (and therefore acts downstream)? Perhaps it is worth doing bacterial two-hybrid assays to detect interactions between DevI and Spo0A or other phosphorelay components?

All excellent suggestions and feasible approaches. There are several other useful approaches that we are also exploring.

The fitness values for ICEBs1+ cells in biofilms without sporulation are reported in text only (subsection “ICEBs1 confers a selective advantage in biofilms without sporulation”). I recommend they be incorporated into Figure 1 using the same format for the data already present.

Done.

Reviewer #4 (Recommendations for the authors):I have only a few questions and suggestions for the authors to consider.– Since eLife is targeted to a broad audience, I think that the detailed discussion of ICE biology and other aspects of the lengthy Introduction would be more digestible if accompanied by a model figure pointing out the key players in ICE function and regulation.

We now include a figure (Figure 1) that we hope better orients readers to ICE biology.

– In the data reported in Figure 4, I was a little surprised that Δ-sncO was less competitive than wild type – I would expect that the convergent transcription of this locus might antagonize transcripts from Pxis, reducing dev1 expression, so I would expect a stronger effect of Dev1 in the deletion of sncO relative to wild type – (maybe the 9.3 vs. 13.8 numbers are really not different, but the authors should comment)

Good eyes! The product of *sncO* may be responsible for a small selective advantage. We do not know the extent to which transcription from P*sncO* interferes with transcription from P*xis*.

– Over-expression of Dev1 from a constitutive promoter clearly had an effect of increasing fitness of the host cell; given the extensive tools available in B subtilis, it would be interesting to use a promoter whose expression can be varied by inducer levels to get an idea of the amount of Dev 1 protein required to inhibit Spo0A – they could start with the lac reporter and then proceed to sporulation/biofilm assays – maybe this is for the next paper, but it is an interesting question related to mechanism.– Finally, the Losick lab published work several years ago on B subtilis cannibalism, where a subfraction of developing cells kill siblings, allowing the killers to obtain additional nutrients and delay sporulation (if I recall correctly). Does the Dev1 system fit into the biology of cannibalism? Authors might comment in the Discussion.

We think that the DevI system could impinge on cannibalism as cannibalism is regulated, in part, by Spo0A. However, the effects of DevI on phenotypes measured here are probably not due to cannibalism. We included a brief mention of this in the Discussion section. DevI-mediated sporulation delay is likely a distinct mechanism from cannibalism, as cannibalism is under the control of Spo0A and the subpopulation of cannibal cells was shown to overlap with matrix-producing cells. Therefore, cells in which DevI is active are less likely to be cannibal cells.